# Synthesis and Biological Activity of Myricetin Derivatives Containing Pyrazole Piperazine Amide

**DOI:** 10.3390/ijms241310442

**Published:** 2023-06-21

**Authors:** Fang Liu, Xiao Cao, Tao Zhang, Li Xing, Zhiling Sun, Wei Zeng, Hui Xin, Wei Xue

**Affiliations:** National Key Laboratory of Green Pesticide, Key Laboratory of Green Pesticide and Agricultural Bioengineering, Ministry of Education, Center for R&D of Fine Chemicals of Guizhou University, Guiyang 550025, China; gs.liuf20@gzu.edu.cn (F.L.); gs.xcao20@gzu.edu.cn (X.C.); gs.zt21@gzu.edu.cn (T.Z.); gs.lxing21@gzu.edu.cn (L.X.); gs.zlsun22@gzu.edu.cn (Z.S.); gs.wzeng22@gzu.edu.cn (W.Z.); gs.hxin22@gzu.edu.cn (H.X.)

**Keywords:** pyrazole, piperazine, myricetin, antifungal activity, SDH, *P. capsici*, *Phomopsis* sp.

## Abstract

In this paper, a series of derivatives were synthesized by introducing the pharmacophore pyrazole ring and piperazine ring into the structure of the natural product myricetin through an amide bond. The structures were determined using carbon spectrum and hydrogen spectrum high-resolution mass spectrometry. Biological activities of those compounds against bacteria, including *Xac* (*Xanthomonas axonopodis pv. Citri*), *Psa* (*Pseudomonas syringae pv. Actinidiae*) and *Xoo* (*Xanthomonas oryzae pv. Oryzae*) were tested. Notably, **D6** exhibited significant bioactivity against *Xoo* with an EC_50_ value of 18.8 μg/mL, which was higher than the control drugs thiadiazole-copper (EC_50_ = 52.9 μg/mL) and bismerthiazol (EC_50_ = 69.1 μg/mL). Furthermore, the target compounds were assessed for their antifungal activity against ten plant pathogenic fungi. Among them, **D1** displayed excellent inhibitory activity against *Phomopsis* sp. with an EC_50_ value of 16.9 μg/mL, outperforming the control agents azoxystrobin (EC_50_ = 50.7 μg/mL) and fluopyram (EC_50_ = 71.8 μg/mL). In vitro tests demonstrated that **D1** possessed curative (60.6%) and protective (74.9%) effects on postharvest kiwifruit. To investigate the active mechanism of **D1**, its impact on SDH activity was evaluated based on its structural features and further confirmed through molecular docking. Subsequently, the malondialdehyde content of **D1**-treated fungi was measured, revealing that **D1** could increase malondialdehyde levels, thereby causing damage to the cell membrane. Additionally, the EC_50_ value of **D16** on *P. capsici* was 11.3 μg/mL, which was superior to the control drug azoxystrobin (EC_50_ = 35.1 μg/mL), and the scanning electron microscopy results indicated that the surface of drug-treated mycelium was ruffled, and growth was significantly affected.

## 1. Introduction

Phytopathogenic fungi and bacteria pose significant hazards to crop growth and the quality of agricultural products, causing substantial economic losses [1,2]. Moreover, the prolonged utilization of conventional fungicides in the market has led to drug resistance of plant pathogens, resulting in recurring bacterial and fungal diseases in most plants [3,4,5,6,7]. In recent years, there has been a heightened emphasis on the ecological environment by the government, prompting researchers in this field to search for efficient, environmentally friendly, and non-polluting agricultural fungicides [8,9].

Flavonoids are a class of plant secondary metabolites found in various plant species, exhibiting complex and distinct structural types. Flavonoids, as known as a natural substance, possess a variety of beneficial physiological activities and are considered safe and widely available. Myricetin is a flavanol compound extracted from the bark and leaves of the Myricaceae plant. It is also found in many edible plants, including fruits and vegetables, as well as many herbs and teas [10,11,12,13,14]. Chemically, myricetin is referred to as 3,5,7,3′,4′,5′-hexahydroxyflavone and belongs to the class of myricetin flavonoids. It has various biological activities such as antibacterial [15], antiviral [16,17], antioxidant [18] and hypoglycemic [19]. In addition, due to its planar structure and multiple phenolic hydroxyl groups, myricetin faces challenges in its poor water solubility and stability. These limitations, which affect its bioavailability and routes of administration, pose significant hurdles to developing myricetin as a pharmaceutical drug. Therefore, finding solutions to enhance the solubility of myricetin and improve its utilization will also be an important area of future research. The structural formula of myricetin is depicted in Figure 1.

With the development of pesticide research, researchers have turned their attention to the direction of natural products in search of green and non-polluting pesticides [20]. Our group discovered myricetin as a promising compound in natural products. We studied myricetin due to its diverse biological activities and incorporated various active heterocyclic groups into its structure. As a result, we obtained compounds that demonstrated significant pesticide activities, including antibacterial and antiviral properties. For example, Jiang [21,22] synthesized compound **1** with an EC_50_ value of 1.58 μg/mL for *Xanthomonas oryzae pv. Oryzae*, compound **2** with an EC_50_ value of 1.1 μg/mL against *Xanthomonas axonopodis pv. Citri*. Chen [23] synthesized compound **3** with an EC_50_ value of 8.2 μg/mL against *Xanthomonas oryzae pv. Oryzae*. Additionally, Peng [24] synthesized compound **4**, demonstrating curative and protective activities against TMV with EC_50_ values of 195.2 and 189.9 μg/mL, respectively. These findings have laid the foundation for discovering more efficient and environmentally friendly botanical pesticides (Figure 2).

Amides contain a unique -C(=O)-NH structure, where the oxygen can form hydrogen bonds with the ubiquinone binding site in succinate dehydrogenase (SDH), and the aromatic ring at the amino terminus binds to the Q site through hydrophobic and π-π interaction, resulting in good biological activity. Amides find extensive applications in pesticides and pharmaceuticals [25]. Among the amide fungicides, succinate dehydrogenase inhibitors (SDHI) are a class of fungicides with novel mechanisms of action, high efficiency and low toxicity, and significant development potential [26].

Pyrazoles are a class of nitrogen-containing five-membered heterocyclic compounds with a wide range of insecticidal [27,28], fungicidal [29,30] and herbicidal [31] activities. pyrazole-4-carbonyl compounds have exhibited good fungicidal activities, making them a hot structure in the research and development of agricultural fungicides [32]. Recent surveys indicated that among the 24 SDHI fungicides on the market, 12 are amide-based fungicides bearing pyrazole heterocycles in their acid components. This highlights the significant role of the pyrazole structure as a crucial pharmacophore in this class of fungicides. Notable commercialized examples of such fungicides include fluxapyroxad, pydiflumetofen, etc., all of which possess broad spectra and excellent antifungal activities (Figure 3) [33,34].

Piperazine is an *N*-heterocyclic basic group that is easy to form multiple hydrogen bonds or ionic bonds. In medicinal chemistry, adjusting the physicochemical properties of compounds can enhance their water solubility, basicity, and biological activities [35]. Piperazine derivatives have attracted considerable interest in medicinal chemistry due to their therapeutic activities, including anti-mycobacterial, antibacterial, antiviral, antifungal, and antitumor effects [36,37,38,39]. In addition, piperazine derivatives have shown promising activities in combating phytopathogenic bacteria, making them valuable subjects for research [40].

Therefore, we introduced the pyrazole amide structure and the piperazine ring into the skeleton of myricetin to synthesize 19 derivatives, aiming to improve the biological activity of myricetin derivatives. The design strategy of the target compound is shown in Figure 4. In addition, their biological activities were evaluated, and the preliminary mechanism of compounds exhibiting excellent biological activity was explored.

## 2. Result and Discussion

### 2.1. Chemistry

The synthetic route of title compounds **D1**–**D19** is illustrated in Figure 1, based on the synthesis method outlined in reference [41,42,43,44,45,46]. The intermediate **a** was synthesized using substituted phenylhydrazine, ethyl acetoacetate or ethyl benzoylacetate, DMF (*N*,*N*-Dimethylformamide) and POCl_3_ (phosphorus oxychloride). Subsequently, intermediat **a** reacted with potassium permanganate (KMnO_4_) under heating conditions, and intermediate **b** was obtained by adjusting pH. Further, Intermediate **b** was reacted with HATU (2-(7-Azabenzotriazol-1-yl)-*N*,*N*,*N*′,*N*′-tetramethyluronium hexafluorophosphate), DIPEA (*N*,*N*-Diisopropylethylamine) and *N*-BOC-piperazine (tert-Butyl 1-piperazinecarboxylate) to synthesize intermediate **c** at room temperature. Intermediate **d** was obtained by adding methanol and HCl solution to the reaction bottle containing intermediae **c**. After adjusting the pH to neutral, DMF, K_2_CO_3_ and intermediat **e** were added to synthesize the target compounds **D1**–**D19**, and their structures were confirmed by ^1^H NMR, ^13^C NMR, and HRMS data; the spectra information (Appendix A) are provided in the Appendix A.

### 2.2. In Vitro Antibacterial Bioassay

The turbidimetric method was used to detect the inhibitory effects of **D1**–**D19** on the plant pathogenic bacteria *Psa*, *Xac* and *Xoo*. The results are shown in Table 1. Compound **D6** demonstrated an impressive 86.5% inhibition of *Xoo* at 100 μg/mL, superior to the commercial drugs thiodiazole-copper (85.8%) and bismerthiazol (78.2%). At the concentration of 50 μg/mL, the inhibition of **D6** against *Xoo* was 76.7%, outperforming thiodiazole-copper (62.1%) and bismerthiazol (57.4%). In addition, at 100 μg/mL, **D6** (49.7%), **D9** (39.5%) and **D13** (44.2%) showed higher inhibition against *Xac* than the control drugs, bismerthiazol (35.6%) and myricetin (22.9%).

Table 1 data shows that when n = 3, R_1_ = −CH_3_, the series of compounds have better inhibitory activity against *Psa*, such as **D1**, **D3**, **D5**, **D9**. In addition, the inhibitory activity of D5 (42.2%) against *Psa* was better than that of **D1** (37%), **D3** (31.6%) and **D9** (38.8%). At the same time, when n = 3, R_1_ = −Ph, the inhibitory activity of compound **D6** against *Xac* (49.7%) and *Xoo* (86.5%) was higher. In summary, it is shown that when n = 3, the substituent −Cl on the benzene ring can improve the antibacterial activity of the compound. Subsequently, the EC_50_ values of **D6**, which had excellent inhibitory activity against *Xoo*, were further evaluated, and the results are presented in Table 2. The results revealed that **D6** had an EC_50_ value of 18.8 μg/mL, which was significantly higher than that of the commercial agents thiodiazole-copper (52.9 μg/mL) and bismerthiazol (69.1 μg/mL).

### 2.3. In Vitro Antifungal Bioassay

The inhibitory activities of the compounds against ten plant pathogenic fungi were determined at 100 μg/mL using azoxystrobin as a control agent. The results are presented in Table 3. The inhibitory effects of **D1** (82.3%), **D2** (72.1%), **D3** (80.5%) and **D9** (81.4%) against *Phomopsis* sp. were better than that of commercial drug azoxystrobin (59.5%). Similarly, **D1** (67.9%) displayed a comparable inhibitory effect to the azoxystrobin (72.8%) against *B. dothidea*. Furthermore, **D1** (37.9%) showed a similar inhibitory effect azoxystrobin (45.1%) against *C. gloeosporioides*. And it also had inhibitory activities against *P. capsici* and *N. dimidiatum*, with inhibition rates of 50.4% and 67.4%, respectively. In summary, compound **D1** showed inhibitory activity against various fungi. To further investigate its mechanism of action, fluopyram was selected as the control agent and *Phomopsis* sp. as the test strain. Its biological activity was measured, and the results showed that the inhibitory activity of **D1** (82.3%) against *Phomopsis* sp. was better than that of fluopyram (48.2%).

Table 3 demonstrated that introducing pyrazole piperazine amide can enhance the antifungal activity of myricetin. In this study, we investigated the association of the pyrazole ring with the parent structure of myricetin and the structures of the substituent group on the pyrazole ring of the compound and analyzed their structure-activity relationships. First, the inhibition activity of the compounds was generally higher when n = 3 (n = number of carbons) than n = 4, demonstrating that the inhibition activity of the target compounds generally decreased with the growth of the carbon chain. Also, we found that the inhibition activity of compounds with R_1_ = −CH_3_ was generally superior to compounds with R_1_ = −Ph (phenyl), such as **D1** > **D2**, **D3** > **D4**, **D5** > **D6**, **D7** > **D8**, **D9** > **D10**. In addition, when R_2_ = −Ph, −4-NO_2_-Ph, −4-Cl-Ph and −4-OMe-Ph, it has inhibitory activity against various fungi, such as **D1**, **D3**, **D5**, and **D9**. In addition, from compounds **D16** (72.8%) and **D19** (76.2%), it can be speculated that the inhibition activity may depend on the exposed heterocyclic structure when the heterocyclic group is far away from the parent structure of myricetin. Based on the in vitro antifungal experiment demonstrated that the introduction of the pyrazole piperazine amide group into the myricetin parent structure could generally improve its biological activity. And when n = 3 and R_1_ = −CH_3_ can effectively improve the antibacterial activity of the compound. To verify the antifungal activity of these compounds more accurately and visually, we determined the half-maximal effective concentration values (EC_50_) of the compounds with higher inhibition than the control agent at 100 μg/mL, and the results are shown in Table 4. The results showed that inhibitory activities of **D1** (16.9 μg/mL), **D2** (19.7 μg/mL) and **D9** (20.3 μg/mL) against *Phomopsis* sp. were superior to control drugs azoxystrobin (50.7 μg/mL) and fluopyram (71.8 μg/mL). In addition, the EC_50_ values of **D1**, **D5**, **D9**, **D10** and **D16** showed between 11.3 and 30.2 μg/mL against *P. capsici*, which were higher than azoxystrobin (35.1 μg/mL) and **D16** (11.3 μg/mL) showed the best performance.

### 2.4. Inhibitory Effect of **D1** on *Phomopsis* sp. In Vitro

Fresh, uniformly sized kiwifruits with no surface damage were selected for experimental studies using the agricultural fungicide fluopyram as a controlled drug. **D1** showed better fungicidal activity against *Phomopsis* sp. than the control drug. Therefore, **D1** was chosen to perform the test to verify its in vitro curative and protective activities against kiwifruit inoculated with *Phomopsis* sp. As shown in Table 5 and Figure 5, the treatment of kiwifruits with **D1** displayed 74.9% protective and 60.6% curative effects, superior to the protective and curative effects of fluopyram (41.4 and 32.5%, respectively). Based on these facts, we can conclude that **D1** has good antifungal activity in vitro.

### 2.5. Light Microscope Observation of Compound **D1** on the Hyphae Morphology

After **D1** (12.5 μg/mL) treatment, *Phomopsis* sp. hyphae were observed under a microscope (100 × 1.25), as depictedas in Figure 6. Specifically, compared to the blank control group (0.5% DMSO), the growth of *Phomopsis* sp. Mycelia were inhibited upon being treated with **D1**. Additionally, increased branching, production of short hyphae and uneven thickness distribution were observed.

### 2.6. Inhibitory Effect of **D1** on SDH In Vitro

Since the structure of this series of compounds is similar to that of succinate dehydrogenase inhibitors, to explore whether SDH is the action site of compound **D1**, the following experiments were carried out. The effects of the target compound **D1** and fluopyram on SDH activity were compared under the same conditions. The experiments involved treating the *Phomopsis* sp. with different concentrations of the drugs and incubating them at a constant temperature of 28 °C for 24 h. The results are shown in Figure 7. Compared with the control drug fluopyram, **D1** had a greater effect on SDH activity and could effectively inhibit SDH activity in a concentration-dependent manner. This result indicates that SDH may be one of the action sites of compound **D1**.

### 2.7. Molecular Docking of **D1** to SDH

The molecular docking results are shown in Figure 8. In this figure, the compounds **D1** and fluopyram are embedded in similar binding modes into the active protein pocket of SDH. These conformations interact with the surrounding amino acid residues via carbon-hydrogen, conventional hydrogen, Pi-alkyl, alkyl, and Pi-sigma. **D1** formed three carbon-hydrogen bonds with THR79 (3.30 Å), SER54 (3.04 Å) and LEU51 (4.50 Å) active sites of SDH, respectively, and fluopyram formed a conventional hydrogen bond with GLY71 (2.07 Å). **D1** formed four Pi-alkyl hydrophobic interactions and alkyl interactions with ILE55 (4.83 Å), LEU72 (4.20 Å), LEU75 (4.86 Å) and VAL54 (4.90 Å). Fluopyram formed five alkyl and Pi-alkyl interactions with LEU75 (4.50, 5.00 and 5.00 Å), ILE78 (4.27 Å) and LEU58 (4.11 Å). In addition, **D1** forms four Pi-sigma interactions with multiple amino acid residues of SDH, such as LEU51 (3.92 Å), LEU58 (3.28 Å) and ILE78 (3.58 and 3.61 Å). Fluopyram forms a Pi-sigma interaction with LEU58 (3.65 Å) and a halogen interaction with LEU75 (3.34 Å). In summary, **D1** forms multiple interactions with the amino acid residues of SDH, making it tightly bound to SDH. At the same time, the docking fraction of **D1** was −5.50, which was better than that of fluopyram at −4.55. It is further proved that SDH may be one of the action sites of **D1**. It is proved that **D1** may affect the tricarboxylic acid cycle by combining with SDH, leading to the final death of fungi.

### 2.8. Effect of **D1** on the Cytoplasmic Leakage of *Phomopsis* sp.

To explore other possible modes of action of compound **D1**, the damage degree of mycelium after drug action was determined. Among them, the malondialdehyde (MDA) content can indirectly reflect the degree of tissue peroxidation damage in the membrane of mycelium, and the higher the content, the more serious the damage to the cell membrane. The results are shown in Figure 9. The content of MDA in *Phomopsis* sp. mycelium treated with different concentrations of **D1** (0, 12.5, 25, 50 and 100 μg/mL) gradually increased with increasing concentration and was better than the control group. The results show that **D1** can cause damage to the cell membrane, and the degree of damage increases with the increase in concentration. It indicates that compound **D1** may also damage the cell membrane structure by destroying the permeability of the cell membrane so that the drug penetrates the bacteria and binds to certain enzymes, affecting the enzyme activity and thereby achieving the antibacterial effect.

### 2.9. Effect of Compound **D16** on Mycelial Morphology of P. capsici

Mycelium of *P. capsici* was treated with 100 μg/mL DMSO (A) and **D16** (B). As shown in Figure 10, the results showed that the mycelium had a smooth surface, great growth, and intact structure after DMSO treatment. However, after **D16** treatment, it experienced shrinkage and folding, with shortened and prominent branches. These changes altered the morphology of the mycelium and significantly inhibited its growth.

## 3. Materials and Methods

### 3.1. Instruments and Chemicals

All reagents were purchased from Shanghai Titan Chemical Co., Ltd. (Shanghai, China) and Ptsrti, Ltd. (Chongqing, China). Enzyme activity kits were purchased from Beijing Solarbio Science & Technology Co., Ltd. (Beijing, China). The melting point data were measured by the X-4B melting point instrument (Shanghai INESA Co., Ltd., Shanghai, China) without correction. Spectral data was measured by a 400 NMR spectrometer with dimethylsulfoxide-d6 as the solvent (Bruker, Karlsruhe, Germany). HRMS data were recorded using a hybrid quadrupole mass spectrometer (Thermo Scientific, Waltham, MA, USA). Image data was obtained on the Olympus CX21FS1 microscope (Tokyo, Japan). Scanning electron microscopy (SEM), data were obtained on FEI Nova Nano 450 (Hillsboro, OR, USA). Thin layer chromatography (TLC) analysis was adopted by a WFH-203B ultraviolet analyzer (Shanghai Jingke Industrial Co., Ltd., Shanghai, China).

#### 3.1.1. Synthesis of Intermediate **a**

Substituted phenylhydrazine (10.4 mmol) and 10 mL anhydrous ethanol were stirred and heated in a 50 mL reaction flask to about 50 °C. Ethyl acetoacetate or ethyl benzoylacetate (10.4 mmol) was then added dropwise. The reaction was refluxed for 5 h to complete, and the solvent was evaporated under reduced pressure to obtain pyrazolone. Take another 100 mL reaction flask, add DMF (29.63 mmol) was added, ice bath to 0 °C, POCl_3_ (70.26 mmol) was slowly added and stirred for 20 min. The above pyrazolone compound was then added slowly. The mixture warmed up to 85 °C and refluxed for 5 h. The progress of the reaction was monitored using TLC (Ethyl acetate: Petroleum ether = 2:1). After cooling to room temperature. The reaction mixture was slowly poured into ice water with constant stirring to accelerate heat dissipation. It was then cooled and left to stand, and intermediate **a** was obtained by filtration, washing, and drying, with a yield of 32–92%.

#### 3.1.2. Synthesis of Intermediate **b**

The intermediate **a** (7.07 mmol) was taken in a 250 mL reaction bottle, and then KMnO_4_ solution (9.2 mmol) was added. The reaction was heated to 80 °C and refluxed for about 8 h. TLC (ethyl acetate: petroleum ether = 2:1) was monitored until the intermediate **a** was completely consumed. After the reaction was cooled to room temperature, the pH was adjusted to alkaline with freshly prepared 10% KOH solution. The insoluble material was removed through filtration, and the resulting transparent filtrate was collected. The pH of the filtrate was then adjusted to acidic by using 10% HCl. The precipitated solid was filtered and dried to obtain intermediate **b**. The yield was 36–91%.

#### 3.1.3. Synthesis of Intermediate **c**

The intermediate **b** (5.52 mmol) was taken in a 100 mL round bottom flask, and mixed with 20 mL DCM (dichloromethane), HATU (6.63 mmol), and DIPEA (8.29 mmol), stirred for 30–45 min at room temperature. Subsequently, N-BOC-piperazine (6.08 mmol) was added and reacted overnight. Then extracted with DCM, the lower extract was collected, the solvent was removed by distillation under reduced pressure, and intermediate **c** was purified by column chromatography with a 14–86% yield.

#### 3.1.4. Synthesis of Intermediate **d**

Weigh intermediate **c** (3.61 mmol) in a 100 mL reaction flask. Add 20 mL of methanol to the flask, slowly add HCl (18 mmol) and react at room temperature until complete. Then, distil under reduced pressure several times to remove HCl and solvent to obtain intermediate **d**. The intermediate **d** was adjusted to neutral pH and entered the next reaction directly.

#### 3.1.5. Synthesis of Intermediate **e**

2.5 g (6.4 mmol) of methylated myricetin and 2.67 g (19.3 mmol) of anhydrous K_2_CO_3_ were taken in a 100 mL reaction flask. 50 mL of DMF was added and stirred for about 30 min at room temperature. Subsequently, 1.96 mL (19.3 mmol) of 1,3-dibromo butane was added dropwise and stirred for 10 h at room temperature. Then the reaction mixture was dispersed in 100 mL of distilled water, leading to the precipitate of a white solid. The solid was then filtered, dried, and poured into a mixed solvent reaction vial containing 60 mL (petroleum ether: ethyl acetate = 1:3, *V/V*). The mixture was stirred for 4–5 h at room temperature, and then filtered and dried to obtain intermediate **e**. The yield was 72%.

#### 3.1.6. Synthesis of Intermediate **D1**–**D19**

DMF (20 mL) and anhydrous K_2_CO_3_ (17.68 mmol) were added to the reaction flask in Section 3.1.4 and then stirred at room temperature for about 45 min. The intermediate **e** (2.95 mmol) was added. And the reaction was carried out at 100 °C for about 5–7 h. After the complete reaction, the reaction system was slowly dispersed in 100 mL of ice water and extracted with DCM (3 × 20 mL). The crude product was obtained by distillation under reduced pressure to remove the solvent and finally purified by column chromatography (ethyl acetate:methanol = 80:1–60:1, *V/V*) to obtain the target compounds **D1**–**D19**, in 4–64% yield.

### 3.2. Antibacterial Activity Bioassay In Vitro

According to the literature [47], the in vitro antibacterial activity of compounds **D1**–**D19** against three plant pathogenic bacteria was determined by the turbidimetric method. *Xanthomonas oryzae pv. Oryzae* (*Xoo*), *Xanthomonas axonopodis pv. Citri* (*Xac*) and *Pseudomonas syringae pv. Actinidiae* (*Psa*) were used as test strains. The prepared NB (Nutrient Broth) medium was sterilized at 121 °C for 20 min. Then the compounds to be tested were dissolved in DMSO and mixed with NB medium to obtain a final concentration of 100 µg/mL. The mixture was added to a 96-well plate, followed by the addition of the prepared bacterial culture (DMSO mixture containing bacteria was used as the negative control, and thiramycin and chlorothalonil were used as the positive control). Another 96-well plate (NB medium with drug and no bacteria) was used as a blank control. The two well plates were sealed and incubated in a constant temperature shaker until the OD_595 nm_ value of the negative control was 0.6–0.7. The OD_595 nm_ values of all bacterial cultures were measured. Triplicates were set for each treatment, and each experiment was repeated three times. The formula was calculated as follows:$$I\left(\%\right)=\left(C-T\right)/C\times 100\%$$

I—inhibition rate

C—OD_595 nm_ (control medium after correction)

T—OD_595 nm_ (drug-containing medium after correction)

According to the above method, the inhibition rates of 100, 50, 25, 12.5, and 6.25 μg/mL were tested. The inhibition rate was converted into probability value (y), and the concentration of the drug solution was converted into logarithmic value (x). The data were processed by Excel software (2021) to obtain the virulence regression equation (y = ax + b) and correlation coefficient (R) to calculate the half maximal effective concentration (EC_50_) of the excellent compound against the pathogen.

### 3.3. Scanning Electron Microscope (SEM) Observation

To further investigate the mechanism of action of these compounds, scanning electron microscopy experiments were carried out with reference to the literature [48,49]. The mycelium treated with **D16** (*P. capsici*) was washed with 0.1 mol/L phosphate buffer (pH = 7.2), fixed in 2.5% glutaraldehyde overnight (4 °C), dehydrated with gradient ethanol, and finally fixed in tert-butanol for 10 min, freeze-dried for about 3 h. The Specimens have been loaded on SEM stubs, sputter-coated with gold, and observed under SEM.

### 3.4. Antifungal Activity Bioassay In Vitro

According to the method reported in the literature, the antifungal activities of the compounds were evaluated with the mycelial growth rate method [50,51,52]. *Phytophthora capsici (Pc^1^)*, *Phomopsis* sp*. (Ps)*, *Botryosphaeria dothidea (Bd)*, *Botrytis cinerea (Bc)*, *Fusarium graminearum (Fg)*, *Fusarium dimerum (Fd)*, *Colletotrichum gloeosporioides (Cg)*, *plectosphaerella cucumerina (Pc)*, *Sclerotinia sclerotiorun (Ss)* and *Neoscytalidium dimidiatum (Nd)* were used as test strains. Firstly, the prepared PDA (Potato Dextrose Agar) medium was sterilized at 121 °C for 20 min. DMSO (0.5%) was the blank control, and azoxystrobin and fluopyram as a positive control. Then the compound to be tested was dissolved in DMSO to make a solution. The solution was then added to the PDA medium to a final concentration of 100 µg/mL. The mixture was thoroughly shaken well and poured into a sterile petri dish. A 5-mm diameter sterile punch was used to punch holes at the edge of the newly activated strains. Subsequently, the blocks were picked with a sterile needle and placed in the center of the drug-containing medium in the above spare petri dishes. Triplicates were set for each treatment, and each experiment was repeated three times. When the mycelium of the blank control group grew until it spread all over the petri dish, the diameter of mycelium growth was measured by the crossover method. The calculation formula was as follows:$$I\left(\%\right)=\left(C-T\right)/\left(C-5\right)\times 100\%.\;$$

“I”—inhibition rate; “C”—blank control; “T”—drug treatment.

### 3.5. In Vitro Antifungal Experiment of **D1** on Kiwifruit

In vitro, bioactivity assay tests were conducted according to the references [53,54]. The in vitro control effect of compound **D1** against *Phomopsis* sp. was determined using the kiwifruit variety “Miliang No 1” as experimental material and the fluopyram as a control drug. To study the curative and protective effects of **D1** on kiwifruit, fresh, uniformly sized kiwifruit fruits with undamaged surfaces were selected for the experimental study, disinfected by immersion in 1% NaClO solution and washed with sterile water. Then the surface water was wiped with filter paper. To evaluate the protective activity, the fruit surface was punched holes (5-mm), and then the prepared **D1** (200 μg/mL) was sprayed uniformly on the surface of kiwifruit fruit, with aqueous DMSO (1%) solution as a blank control. And 5-mm agar blocks containing mycelium were placed on the fruit perforations after 24 h in the incubator. To evaluate the curative activity, agar blocks containing mycelium were inoculated at the fruit perforations and placed in the incubator for 24 h. **D1** (200 μg/mL) was sprayed evenly on the surface of the fruit. The fruits were incubated in an incubator (25 °C, 85% relative humidity) for 96 h after inoculation with pathogens, and then the diameter of the spots was measured. Calculations were performed as follows:C (%) = [(A_CK_ − A_1_)/(A_CK_ − 5)] × 100.

C—control effect;

A_CK_—lesion diameter of the blank control group;

A_1_—lesion diameter after compound treatment.

C represents the control effect (%);

There were three parallel sets for each concentration, and the experiments were repeated at least twice.

### 3.6. In Vitro SDH Inhibition Assay

The succinate dehydrogenase assay kit used in the experiment was procured from Beijing Solarbio Technology Co, Beijing, China [55]. It was used to determine the inhibition of the target compound **D1** against SDH in vitro, and the commercial SDHI fluopyram was used as a positive control. The newly activated bacteria were put into the sterilized PDB (Potato Dextrose Broth) medium and incubated in a constant temperature shaker (25 °C, 180 rpm/min). The drug solution was added to the above medium containing mycelium to a final concentration of 100, 50, 25 and 12.5 μg/mL and incubated in a shaker for 24 h. The mycelium was collected, freeze-dried, and ground in liquid nitrogen. Weigh 0.1 g of mycelium treated with different concentrations, assay according to the kit's instructions, set up three parallels each time, and repeat three times.

### 3.7. Molecular Docking

To further investigate whether SDH is a potential target for action, we performed molecular docking of **D1** and commercially available SDHI fluopyram with SDH (PDB:2FBW) [56,57]. The experimental results were further validated at the molecular level.

### 3.8. Determination of Malondialdehyde Content

To further study the mechanism of action of these compounds, we conducted a malondialdehyde content test. Malondialdehyde (MDA) is one of the main products of membrane lipid peroxidation. Its content is usually used as an indicator of lipid peroxidation, reflecting the degree of cell membrane damage [54,58]. The MDA content detection kit used in the experiment was purchased from Beijing Soleibao Technology Co. The young viable *Phomopsis* sp. strains were selected in a sterilized PDB medium and incubated in a constant temperature shaker for 48 h. The strains were incubated in a **D1** solution containing concentrations (0, 25, 50, 100, 200 μg/mL) for 24 h. Rinse the medium with sterile water, then collect the mycelium and freeze-dried. 0.1 g of mycelium treated with different concentrations of the solution was weighed separately and tested according to the instructions of the kit. Finally, the absorbance of each sample was measured at 532 nm and 600 nm.

## 4. Conclusions

We designed and synthesized 19 myricetin derivatives containing pyrazole-piperazinamide in this study and conducted their biological activities tests. The in vitro antibacterial experiments revealed that **D6** exhibited good inhibitory activity against *Xoo*, with an EC_50_ value of 18.8 μg/mL. In vitro antifungal experiments demonstrated that **D16** displayed potent inhibitory activity against *P. capsici*, with an EC_50_ value of 11.3 μg/mL. The SEM analysis showed that **D16** induced folding and curing and inhibited the growth of the *P. capsici* mycelium. Moreover, compound **D1** exhibited remarkable inhibitory effects against *Phomopsis* sp., with an EC_50_ value of 16.9 μg/mL. The in vitro protective effect of **D1** against *Phomopsis* sp. was 74.9%, and the curative effect was 60.6%. Microscopic experiments demonstrated that **D1** inhibited mycelial growth. And **D1** can effectively inhibit the activity of SDH, and molecular docking studies confirmedits strong binding affinity with SDH protein. Additionally, **D1** increased the malondialdehyde level, leading to cell membrane damage. Based on these findings, it can be concluded that myricetin derivatives containing pyrazole piperazine amide offer new insights and a theoretical basis for developing potential fungicides.

## Data Availability

All data generated in this study is presented in the current manuscript. No new datasets were generated. Data is available upon request from the corresponding author. Informed consent was obtained from all subjects involved in the study.

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
