# Peer review of "Synthesis and Biological Activity of Myricetin Derivatives Containing Pyrazole Piperazine Amide"

_ijms, 2023, doi:10.3390/ijms241310442_

Round 1

Reviewer 1 Report (Previous Reviewer 1)

Manuscript ID: ijms-2451446 

The Article "Synthesis and Biological Activity of Myricetin Derivatives Containing Pyrazole Piperazine Amide" by Xue et al submitted to International Journal of Molecular Sciences.

The authors have improved the manuscript in this resubmission of the article.

Author Response

Dear editor and reviewers:

Thank you very much for your letter and the comments about our paper entitled “Synthesis and Biological Activity of Myricetin Derivatives Containing Pyrazole Piperazine Amide” (ID: ijms-2451446). Those comments are all valuable and very helpful for revising and improving our paper, as well as the important guiding significance to our researches.

We have studied comments carefully and have made correction which we hope meet with approval. These revisions marked with red font in the manuscript. Please find the revised version, which we would like to submit for your kind consideration. We would like to express our great appreciation to your comments on our paper.

Reviewer 2 Report (New Reviewer)

REVIEW ON ijms-2451446

In this paper, Wei Xue et al reported the synthesis of myrcetin derivatives and their biological evaluation against a panel of phytopathogenic microbial strains. From my point of view, the design of experiment is clearly stated and understandable. The targeted compounds are obtained in a 6 steps process coupling myrcetin (natural compound) with piperazine/pyrazole moieties. A series of 19 derivatives was evaluated and gave some interesting results in terms of activity. The chemistry used classical synthetic processes and is well described.

Overall, this work is worth to be published, due to the gathering of interesting data, mixing design, docking, synthesis and biological evaluation. However, from my point of view, some points are to be improved and addresses by the authors before publication:

·  The authors stated that pyrazole structure is a crucial pharmacophore but do give only examples were the pyrazole exhibited N-methylation and modification by an halogenated moiety. Could they discuss about the real pharmacophore? If the CHF2 group was deleted from Fluxapyroxad (in Figure 3), is the resulting compound still active?

 ·In 2.1 please correct DMF (N,N-dimethylformam) by DMF (N,N-dimethylformamid)

 ·       L135: please add a blank line after Table 1

 ·       How were the reference compounds selected? Please explain.

 ·       Paragraphs 2.2 and 2.3.: please discuss the structure/activity relationships of the compounds. For the moment, the results are only summarized but not fully analyzed from my point of view. In particular, please analyze the impact of the chemical modifications on activity.  

 ·       2.4. The authors should detail what is the “protective” and the “curative” assays to help clarify the interest of the compounds.

 ·       L195: please add a blank “depicted as”

 ·       2.5. Light microscope observation; please correct the typo

 ·       2.5. Please give some interpretation of these data.

 ·       2.8. Malondialdehyde seems to be the chemical component involved in the destruction of the cell membrane. Could you please give us some data where malondialdehyde has been tested alone as permeabilizer? More specifically, please precise at which concentration malonadialdehyde exerts its activity. Is that concentration comparable with the one observed in your assays?

 ·       The authors evaluated their products as potential pesticides ; could they discuss about the feasibility of entering the market for such biosourced compounds, in particular from a cost point of view ?

 ·       On point which is mentioned by the authors is the low solubility of myrcetin. Did they evaluate the impact of their chemical modifications on solubility? It should be of interest for future applications.

Author Response

Dear editor and reviewers:

Thank you very much for your letter and the comments about our paper entitled “Synthesis and Biological Activity of Myricetin Derivatives Containing Pyrazole Piperazine Amide” (ID: ijms-2451446). Those comments are all valuable and very helpful for revising and improving our paper, as well as the important guiding significance to our researches.

We have studied comments carefully and have made correction which we hope meet with approval. These revisions marked with red font in the manuscript. Please find the revised version, which we would like to submit for your kind consideration. We would like to express our great appreciation to your comments on our paper. The main corrections in the paper and the responds to the reviewer’s comments are as following:  

Point 1: The authors stated that pyrazole structure is a crucial pharmacophore but do give only examples were the pyrazole exhibited N-methylation and modification by an halogenated moiety. Could they discuss about the real pharmacophore? If the CHF2 group was deleted from Fluxapyroxad (in Figure 3), is the resulting compound still active?

Response 1: Thank you for your suggestion. The activity of pyrazole compounds is determined by the combination of pyrazole and the substituents on the pyrazole ring. Nowadays, there are many commercial fungicides containing pyrazole structure on the market, which also fully proves that pyrazole is active, and the substituents will also enhance the antibacterial activity of the whole compound. However, the determination of the true pharmacophore has not been mentioned in detail in the references. This article only selects the pyrazole heterocyclic structure through a large number of literature records to study its activity after binding to the lead compound.

Point 2: In 2.1 please correct DMF (N,N-dimethylformam) by DMF (N,N-dimethylformamid).

Response 2: Thank you for your suggestion. DMF (N, N-dimethylformam) has been modified to DMF (N, N-dimethylformamid),please refer to the revision details of section 2.1 in the revised version.

Point 3: please add a blank line after Table 1.

Response 3: Thank you for your suggestion. A blank line has been added after Table 1. Please refer to the modified details in the revised version.

Point 4: How were the reference compounds selected? Please explain.

Response 4: Thank you for your suggestion. The reason why thiodiazole-copper and bismerthiazol were selected as antibacterial reference drugs : thiodiazole-copper is a thiazole organic copper fungicide, which has a special effect on the prevention and treatment of crop bacterial diseases. It has the characteristics of novel structure, advanced dosage form, good internal conductivity, low toxicity and safety. The bismerthiazol is mainly used to control plant bacterial diseases. It has a good control effect on X.oryzae pv.oryzae, X.oryzae pv.oryzicola and P. syringae pv. actinidiae. It is an internal fungicide with good curative and protective effects and has a good control effect on bacterial diseases. Azoxystrobin is a methoxyacrylate fungicide with high efficiency and broad spectrum. It has good activity against almost all fungal diseases. Fluopyram is a succinate dehydrogenase inhibitor, which is an effective fungicide. It is safe to crops and the environment, has good protection and strong permeability, and has a good inhibitory effect on various main forms of pathogenic fungi. The above control agents are often used in our research group and related research.

Point 5: Paragraphs 2.2 and 2.3.: please discuss the structure/activity relationships of the compounds. For the moment, the results are only summarized but not fully analyzed from my point of view. In particular, please analyze the impact of the chemical modifications on activity.

Response 5: Thank you for your suggestion. In section 2.2, the structure-activity relationship analysis was supplemented, such as : “Table 1 data shows that when n=3, R1=-CH3, the series of compounds have better inhibitory activity against Psa, such as D1, D3, D5, D9. In addition, the inhibitory activity of D5 (42.2%) against Psa was better than that of D1 (37%), D3 (31.6%) and D9 (38.8%). At the same time, when n=3, R1=-Ph, the inhibitory activity of compound D6 against Xac (49.7%) and Xoo (86.5%) was higher. In summary, it is shown that when n=3, the substituent -Cl on the benzene ring can improve the antibacterial activity of the compound”; In section 2.3, we added the relevant content, such as : “In addition, when R2=-Ph, -4-NO2-Ph, -4-Cl-Ph and -4-OMe-Ph, it has inhibitory activity against a variety of fungi, such as D1, D3, D5, D9.” Please refer to the modified details in the revised version.

Point 6: 2.4. The authors should detail what is the “protective” and the “curative” assays to help clarify the interest of the compounds.

Response 6: Thank you for your suggestion. The “protective” determination refers to: In order to protect plants, fungicides are used to eliminate pathogenic fungi or prevent the invasion of pathogenic fungi before the plant is not diseased. That is, fungicides establish a chemical barrier between pathogens and plants, so that pathogenic fungi cannot successfully infect plants. The “curative” determination refers to: After the plant is infected or diseased, fungicides are applied to the plant to relieve the relationship between the pathogenic fungi and the host or prevent the development of the disease, so that the plant can restore health.

Point 7: L195: please add a blank “depicted as”.

Response 7: Thank you for your suggestion. We have modified it as required, Please refer to the modified details in the revised version.

Point 8: 2.5. Light microscope observation; please correct the typo.

Response 7: Thank you for your suggestion. “2.5. light microscope“ has been modified to “Light Microscope Observation”, please refer to the revision of the 2.5 part of the details.

Point 9: 2.5. Please give some interpretation of these data.

Response 9: Thank you for your suggestion. 100 × 1.25 : Indicates that the objective magnification is 100 times and the numerical aperture is 1.25.

Point 10: 2.8. Malondialdehyde seems to be the chemical component involved in the destruction of the cell membrane. Could you please give us some data where malondialdehyde has been tested alone as permeabilizer? More specifically, please precise at which concentration malonadialdehyde exerts its activity. Is that concentration comparable with the one observed in your assays?

Response 10: Thank you for your suggestion. In this paper, only the preliminary mechanism of action was explored. Due to the limitation of experimental conditions, more in-depth research was not carried out. Only the malondialdehyde kit was used to detect the change of malondialdehyde content in mycelium after treatment with different concentrations of drugs, so as to judge the damage of compounds to fungi.

Point 11: The authors evaluated their products as potential pesticides ; could they discuss about the feasibility of entering the market for such biosourced compounds, in particular from a cost point of view ?

Response 11: Thank you for your suggestion. Firstly, myricetin is a natural product with a wide range of sources, and has a variety of pharmacological activities, which is of research significance. Research based on natural products is regarded as one of the important ways to create and develop new pesticides. At the same time, there are many widely used natural product-derived pesticides on the market, such as pyrethroid pesticides. In addition, it is conducive to the discovery of new pesticide targets, thus providing a biological basis for the creation of pesticides with new mechanisms of action ; for natural product derivatives with good activity, it is possible to enter the market in the future by continuously optimizing the synthesis steps to save costs.

Point 12: On point which is mentioned by the authors is the low solubility of myrcetin. Did they evaluate the impact of their chemical modifications on solubility? It should be of interest for future applications.

Response 12: Thank you for your suggestion. The “has become an important area of research“ of L53 has been modified to “will also be an important area of future research”, Please refer to the modified details in the revised version. 

This manuscript is a resubmission of an earlier submission. The following is a list of the peer review reports and author responses from that submission.

Round 1

Reviewer 1 Report

The title “Synthesis and biological activity of Myricetin derivatives containing Pyrazole Piperazine amide” by Fang Liu et al.

In this manuscript authors provide evidence of activity improvement compared to Myricetin derivative while they have tested biological activity against bacteria and fungi on their Prepared derivatives of pyrazole-piperazine amides.

Based on their observation, the introduction of the piperazine ring improved the solubility problem of myricetin analogues and addition of pyrazole ring enhanced the antibacterial and antifungal activity of prepared derivatives.

Even though earlier seminal research reports are available in the literature, which is relevant, this research article will be a merit in publishing in this journal for readers perspective.

Synthetic protocol for reported molecules is well documented with proper supporting information data (procedure, characterization including melting point, NMR data, mass data). They have provided appropriate references in support of the research article.

Author Response

Dear editor and reviewers:

Thank you very much for your letter and the comments about our paper entitled “Synthesis and Biological Activity of Myricetin Derivatives Containing Pyrazole Piperazine Amide” (ID: ijms-2400908). Those comments are all valuable and very helpful for revising and improving our paper, as well as the important guiding significance to our researches.

We have studied comments carefully and have made correction which we hope meet with approval. These revisions marked with red font in the manuscript. Please find the revised version, which we would like to submit for your kind consideration. We would like to express our great appreciation to your comments on our paper. The main corrections in the paper and the responds to the reviewer’s comments are as following:

We have invited professionals to make extensive English revisions to this article. Please refer to the revised version for details.

Reviewer 2 Report

The authors have synthesized a series of Myrecitin derivatives and shown considerable antibacterial and antifungal properties of certain hits. The results are publishable with the following corrections

1) Provide a water solubility chart of these molecules relative to myrecitin.

2) The mechanism of antifungal activity by D1 is loosely defined in the paper. I would recommend providing various spectroscopic proofs (NMR, IR etc.) for pre and pro in-vivo studies.

3) The authors should not use 'Green pesticide's since the synthesis involves several toxic chemicals.

Have you considered a cocktail of molecules against a series of bacteria and fungi?

Author Response

Dear editor and reviewers:

Thank you very much for your letter and the comments about our paper entitled “Synthesis and Biological Activity of Myricetin Derivatives Containing Pyrazole Piperazine Amide” (ID: ijms-2400908). Those comments are all valuable and very helpful for revising and improving our paper, as well as the important guiding significance to our researches.

We have studied comments carefully and have made correction which we hope meet with approval. These revisions marked with red font in the manuscript. Please find the revised version, which we would like to submit for your kind consideration. We would like to express our great appreciation to your comments on our paper. The main corrections in the paper and the responds to the reviewer’s comments are as following:  

Point 1: Provide a water solubility chart of these molecules relative to myrecitin.

Response 1: Thank you for your suggestion. Due to the limited laboratory conditions, we generally evaluated the dissolution of this series of compounds in 100g water as follows :

compound

Sparingly soluble(0.01g-1g)

D4,D7,D10

Difficultly soluble(<0.01g)

D1,D2,D3,D5,D6,D8,D9,D11,D12,D13,D14,D15,D16,D17,D18,D19

Point 2: The mechanism of antifungal activity by D1 is loosely defined in the paper. I would recommend providing various spectroscopic proofs (NMR, IR etc.) for pre and pro in-vivo studies.

Response 2: Thank you for your suggestion. In this paper, the structure of D1 was determined by 1H NMR, 13C NMR and HRMS. The definition of related mechanisms is supplemented. Please refer to the revision details.

Point 3: The authors should not use 'Green pesticide's since the synthesis involves several toxic chemicals.

Response 3: Thank you for your suggestion. The relevant description has been modified. Please refer to the modified details in the revised version.

Reviewer 3 Report

The authors present a study on the biological activities of myricetin derivatives against bacteria and fungi. This research involved significant dedication and effort. However, the written language lacks quality, and the conclusions are overly assertive. I have the following suggestions for the authors:

Major concerns:

1.       The authors conducted succinate dehydrogenase experiments without establishing a clear correlation between its molecular role and the mode of action of myricetin derivatives.

2.       The authors should understand that results and methods sections are distinct. Section 2.1 should encompass the methodology rather than the results.

3.       Scanning electron microscopy was employed to examine the morphological changes induced by test compounds. However, without tagging the target molecules with identifiable dyes, it may not provide accurate information about the mechanism of action at the molecular level.

4.       The authors observed morphological changes, such as shrinkage and wrinkles, under the scanning electron microscope and prematurely attributed them to the influence of the test molecules. These changes could potentially be artifacts introduced by the pH or osmolarity of the test samples. The authors need to ensure that the morphological changes are not due to changes in the pH and osmolarity. Additionally, presenting high-resolution micrographs would facilitate a more precise evaluation.

5.       The author performed antibacterial activities of their compounds on kiwifruit and erroneously categorized it as in vivo experiments. Although fruit is derived from living organisms, conducting experiments solely on fruit generally falls under in vitro or ex vivo experiments.

6.       The author's basic chemistry knowledge is inadequate, as evidenced by their description of the bond between carbon and hydrogen as an intermolecular bond with a length of 3.30 angstroms, which is implausible. Additionally, they inaccurately describe a hydrogen bond with a length exceeding 4 angstroms and mention pi-pi interactions between linear molecules, which is entirely incorrect.

7.       The author did not mention MDA (malondialdehyde) in detail and its relationship to cell membrane damage.

Minor Concerns:

1.       The authors falsely claim to have determined the structure of myricetin through high-resolution mass spectrometry (HRMS). HRMS is a powerful technique commonly employed for the structural elucidation of small molecules. While HRMS alone may not provide complete and detailed structural determination, it can offer vital information about the molecular formula and fragmentation pattern, aiding in structure determination.

 2.       The author's choice of words appears superficial, non-scientific, and inappropriate in the given context. Several words such as "better inhibition," "novel mechanism" (without detailed explanation), "hot structure," "good biological activity," "better than," "better inhibition," "general antifungal activity," "certain antifungal activity," "normal mycelium," "somewhat inhibited," "precipitation dried by filtration," "react overnight," "general procedure," "until complete," "96 well-plate method," "bacterial culture" (not bacterial solution), "three replicate (instead triplicate)," "on SEM stage" (instead of SEM stub), "sprayed with gold" (instead of sputter coated with gold), "on SEM" (instead of under SEM), "growth rate method," "dried with filter paper" (instead of wiped with filter paper), "perforated with punch" (instead of punched holes), "cultured" (instead of incubated), "fully ground," "cell membrane damage," "poor solubility problem," "against kiwifruit," "folded and curled" should be reconsidered.

 3.       Several abbreviations were used without providing their full forms, such as MDA, microorganism names in Table 1, CK in Figure 7, POCl3, DMF, DCM, HATU, DIPEA, N-BOC, PDA, Ps, and PD.

 4.       The author cited previous studies without providing substantial information about the names of compounds and microorganisms against which biological activities were tested.

 5.       Figures 2 and 3 lack important information such as compound names, figure legends, microorganisms tested, and citations to relevant studies.

 6.       In section 2.3, the author mentions conducting three replicates. It is unclear whether this means performing experiments six times for each group or performing experiments in triplicate. This should be clarified.

7.       The author does not clarify the meaning of 'n' and R1=Ph in section 2.5.

8.       In Section 2.4, the author performed a microscopy examination of mycelium treated with their compound but did not specify the type of microscopy, scale bar, and resolution used.

9.       The reference to a "dark-colored compound" in section 3.1 requires clarification.

 10.   The author did not mention the pH they adjusted to prepare the intermediate compounds.

 11.   The author should have explicitly stated that section 3.1.3 describes the procedure for synthesizing intermediate compounds C and D.

12.   The author did not explain the basis for differentiating compounds D1-D19 in the synthesis procedure.

13.   In section 3.2, the author refers to the bacterial culture as a solution which is inappropriate and, did not specify which compounds were tested.

14.   The author mentioned taking specimens on the SEM stage, spraying them with gold, and observing them on SEM, which is incorrect. Specimens should have been loaded on SEM stubs, sputter-coated with gold, and observed under SEM.

15.   The author did not specify what the blank control refers to in Section 3.4.

16.   In section 3.5, the specific drug tested by the author is not specified. Additionally, the author repeats information that has already been provided in section 3.1.

17.   The author's description of the docking of fluopyram with SDH and D1 may confuse the reader, as it is incorrect or misleading.

18.   The author should clearly indicate the solution used to treat the mycelium and provide an explanation for using two separate frequencies to measure the optical densities.

19.   The author did not explain how they calculated the EC50 values from the optical densities.

20.   The author did not mention the specific mechanistic studies they conducted.

21.   The conclusion section provides a brief summary of the results rather than offering an overall inference or conclusion drawn from the study.

1.       There are multiple sentences that are redundant and challenging to comprehend due to poor writing, such as in line numbers 32, 42-43, 56-57, 103, 122-123, 128-129, 144-147, 237-238, 289-291, 319-321.

2.       The author's choice of words appears superficial, non-scientific, and inappropriate in the given context. Several words such as "better inhibition," "novel mechanism" (without detailed explanation), "hot structure," "good biological activity," "better than," "better inhibition," "general antifungal activity," "certain antifungal activity," "normal mycelium," "somewhat inhibited," "precipitation dried by filtration," "react overnight," "general procedure," "until complete," "96 well-plate method," "bacterial culture" (not bacterial solution), "three replicate (instead triplicate)," "on SEM stage" (instead of SEM stub), "sprayed with gold" (instead of sputter coated with gold), "on SEM" (instead of under SEM), "growth rate method," "dried with filter paper" (instead of wiped with filter paper), "perforated with punch" (instead of punched holes), "cultured" (instead of incubated), "fully ground," "cell membrane damage," "poor solubility problem," "against kiwifruit," "folded and curled" should be reconsidered.

3.       The author's description of the docking of fluopyram with SDH and D1 may confuse the reader, as it is incorrect and misleading.

4.       Random capital letters have been utilized in the text, for example, in line 29.

Author Response

Dear editor and reviewers:

Thank you very much for your letter and the comments about our paper entitled “Synthesis and Biological Activity of Myricetin Derivatives Containing Pyrazole Piperazine Amide” (ID: ijms-2400908). Those comments are all valuable and very helpful for revising and improving our paper, as well as the important guiding significance to our researches.

We have studied comments carefully and have made correction which we hope meet with approval. These revisions marked with red font in the manuscript. Please find the revised version, which we would like to submit for your kind consideration. We would like to express our great appreciation to your comments on our paper. The main corrections in the paper and the responds to the reviewer’s comments are as following:  

Point 1: The authors conducted succinate dehydrogenase experiments without establishing a clear correlation between its molecular role and the mode of action of myricetin derivatives.

Response 1: Thank you for your suggestion. The relationship between the molecular action of succinate dehydrogenase experiment and the mode of action of myricetin derivatives was re-expressed. Please refer to the revision details in Section 2.3.4 of the revised version.

Point 2: The authors should understand that results and methods sections are distinct. Section 2.1 should encompass the methodology rather than the results.

Response 2: Thank you for your suggestion. This part has been re-modified, please refer to the revision details of section 2.1 in the revised version.

Point 3: Scanning electron microscopy was employed to examine the morphological changes induced by test compounds. However, without tagging the target molecules with identifiable dyes, it may not provide accurate information about the mechanism of action at the molecular level.

Response 3: Thank you for your suggestion. The 2.3.7 part of this paper is the scanning electron microscope imaging experiment of the effect of compound D16 on the morphology of Phytophthora capsici. please refer to the revision details of Section 2.3.7. in the revised version.

Point 4: The authors observed morphological changes, such as shrinkage and wrinkles, under the scanning electron microscope and prematurely attributed them to the influence of the test molecules. These changes could potentially be artifacts introduced by the pH or osmolarity of the test samples. The authors need to ensure that the morphological changes are not due to changes in the pH and osmolarity. Additionally, presenting high-resolution micrographs would facilitate a more precise evaluation.

Response 4: Thank you for your suggestion. The whole preparation process of the blank control group and the drug treatment group was the same. The pH of the medium and buffer used in the experiment was the standard value, which could ensure that the change of morphology was not due to the change of pH and osmotic pressure. In addition, the cell membrane was damaged after drug treatment, and there were holes on the surface, which was caused by drug action. Please refer to the modified details in the revised version. High-resolution micrographs has been provided, please refer to the figure in the “Figure and Graphic”.

Point 5: The author performed antibacterial activities of their compounds on kiwifruit and erroneously categorized it as in vivo experiments. Although fruit is derived from living organisms, conducting experiments solely on fruit generally falls under in vitro or ex vivo experiments.

Response 5: Thank you for your suggestion. Although the fruit leaves the fruit tree, it still contains a large number of active substances, such as cells and enzymes, and there are still life activities. Relevant literature also classifies experiments on fruits and leaves as in vivo experiments. Therefore, this paper classifies the antibacterial activity of compounds on kiwifruit as in vivo experiments. Please refer to the modified details in the revised version.

Point 6: The author's basic chemistry knowledge is inadequate, as evidenced by their description of the bond between carbon and hydrogen as an intermolecular bond with a length of 3.30 angstroms, which is implausible. Additionally, they inaccurately describe a hydrogen bond with a length exceeding 4 angstroms and mention pi-pi interactions between linear molecules, which is entirely incorrect.

Response 6: Thank you for your suggestion. The molecular docking part has been re-described. Please refer to the revised version of the 2.3.5. part of the modification details.

Point 7: The author did not mention MDA (malondialdehyde) in detail and its relationship to cell membrane damage.

Response 7: Thank you for your suggestion. “In order to further investigate the mechanism of action of the compounds, we conducted a malondialdehyde content detection test to determine the extent of cell membrane damage by its content” was modified to “”In order to further study the mechanism of action of these compounds, we conducted a malondialdehyde content test. Malondialdehyde (MDA) is one of the main products of membrane lipid peroxidation. Its content is usually used as an indicator of lipid peroxidation, reflecting the degree of cell membrane damage”, please refer to the revision of the 3.8 part of the details.

Minor Concerns:

Point 1: The authors falsely claim to have determined the structure of myricetin through high-resolution mass spectrometry (HRMS). HRMS is a powerful technique commonly employed for the structural elucidation of small molecules. While HRMS alone may not provide complete and detailed structural determination, it can offer vital information about the molecular formula and fragmentation pattern, aiding in structure determination.

Response 1: Thank you for your suggestion. The structures of all compounds were determined by 1H NMR, 13C NMR and HRMS data. Please refer to the revision details and support information in Section 2.1 of the revised version.

Point 2: The author's choice of words appears superficial, non-scientific, and inappropriate in the given context. Several words such as "better inhibition," "novel mechanism" (without detailed explanation), "hot structure," "good biological activity," "better than," "better inhibition," "general antifungal activity," "certain antifungal activity," "normal mycelium," "somewhat inhibited," "precipitation dried by filtration," "react overnight," "general procedure," "until complete," "96 well-plate method," "bacterial culture" (not bacterial solution), "three replicate (instead triplicate)," "on SEM stage" (instead of SEM stub), "sprayed with gold" (instead of sputter coated with gold), "on SEM" (instead of under SEM), "growth rate method," "dried with filter paper" (instead of wiped with filter paper), "perforated with punch" (instead of punched holes), "cultured" (instead of incubated), "fully ground," "cell membrane damage," "poor solubility problem," "against kiwifruit," "folded and curled" should be reconsidered.

Response 2: Thank you for your suggestion. The grammar of the whole article has been modified, and the relevant terms have been replaced and modified according to the review opinions. Please refer to the modified details in the revised version.

Point 3: Several abbreviations were used without providing their full forms, such as MDA, microorganism names in Table 1, CK in Figure 7, POCl3, DMF, DCM, HATU, DIPEA, N-BOC, PDA, Ps, and PD.

Response 3: Thank you for your suggestion. The full forms of all abbreviations is provided in this article. The full name of “MDA” is “malondialdehyde”; The complete form of the bacterial name has been recently supplemented in Table 1.; The CK of Figure 7. has been explained in Section 2.3.3.,  “POCl3, DMF, HATU, DIPEA, N-BOC” have been marked in the first part of the complete form, see Section 2.1.; DCM (dichloromethane) has been marked in Section 3.1.3.; In Section 3.2, we provide the complete form of “NB (Nutrient Broth)”. The complete form of “PDA” is provided in section 3.4.,namely “Potato Dextrose Agar”. The complete forms of Ps are provided in the full text. The complete form of “PD” is “PDB ( Potato Dextrose Broth ) “. Please refer to the modified details in the revised version.

Point 4: The author cited previous studies without providing substantial information about the names of compounds and microorganisms against which biological activities were tested.

Response 4: Thank you for your suggestion. Substantive information on compounds and microbial names for testing biological activity has been provided in the paper. Please refer to the revision details of the introduction (Lines 65-69) in the revised version.

Point 5: Figures 2 and 3 lack important information such as compound names, figure legends, microorganisms tested, and citations to relevant studies.

Response 5: Thank you for your suggestion. The names of the compounds have been provided in Figure 2 and Figure 3, and important information such as microbial names and citations of related studies have been supplemented and modified in the text. In addition, the compounds shown in Figure 3 are all commercially available succinate dehydrogenase inhibitors. Each drug has good inhibitory activity against dozens of plant pathogenic fungi. Therefore, not all listed in the text, please refer to the reference literature in detail. Please refer to the modified details in the revised version.

Point 6: In section 2.3, the author mentions conducting three replicates. It is unclear whether this means performing experiments six times for each group or performing experiments in triplicate. This should be clarified.

Response 6: Thank you for your suggestion. In response to this suggestion, it is clarified in section 3.5 of “Materials and Methods”, as follows “There were three parallel sets for each concentration, and the experiments were repeated at least twice “. Please refer to the modified details in the revised version.

Point 7: The author does not clarify the meaning of 'n' and R1=Ph in section 2.5.

Response 7: Thank you for your suggestion. In section 2.4, we clarify the meaning of n and R1 = Ph, “n = number of carbons “, “R1 = -Ph ( phenyl group )”. Please refer to the modified details in the revised version.

Point 8: In Section 2.4, the author performed a microscopy examination of mycelium treated with their compound but did not specify the type of microscopy, scale bar, and resolution used.

Response 8: Thank you for your suggestion. The microscope type, scale and resolution are provided in section 3.1 and 2.3.3. Please refer to the modified details in the revised version.

Point 9: The reference to a "dark-colored compound" in section 3.1 requires clarification.

Response 9: Thank you for your suggestion. “Dark-colored compound” has been changed to “pyrazolone “. Please refer to the revision details in section 3.1.1 of the revised version.

Point 10: The author did not mention the pH they adjusted to prepare the intermediate compounds.

Response 10: Thank you for your suggestion. The preparation of intermediate compounds by adjusting pH was mentioned in section 3.1.2 and section 3.1.4. Please refer to the modified details in the revised version.

Point 11: The author should have explicitly stated that section 3.1.3 describes the procedure for synthesizing intermediate compounds C and D.

Response 11: Thank you for your suggestion. The section “3.1.4. General Procedure for the Synthesis of Intermediate d” was added. Please refer to the modified details in the revised version.

Point 12: The author did not explain the basis for differentiating compounds D1-D19 in the synthesis procedure.

Response 12: Thank you for your suggestion. The 3.1.1~3.1.6 section of the synthetic process has been modified to illustrate the basis for distinguishing compounds D1-D19. Please refer to the modified details in the revised version.

Point 13: In section 3.2, the author refers to the bacterial culture as a solution which is inappropriate and, did not specify which compounds were tested.

Response 13: Thank you for your suggestion. It has been shown that the tested compounds are D1-D19, and the “solution” is modified to ”culture“. Please refer to the revision details in section 3.2 of the revision.

Point 14: The author mentioned taking specimens on the SEM stage, spraying them with gold, and observing them on SEM, which is incorrect. Specimens should have been loaded on SEM stubs, sputter-coated with gold, and observed under SEM.

Response 14: Thank you for your suggestion. The section 3.3 has been modified. The “A small amount of sample was taken on the sample stage, sprayed with gold and observed on the SEM” was modified to “The have been loaded on SEM stubs, sputter-coated with gold, and observed under SEM”. Please refer to the modified details in the revised version.

Point 15: The author did not specify what the blank control refers to in Section 3.4.

Response 15: Thank you for your suggestion. The section 3.4 has been modified to specify that blank control refers to “0.5 % DMSO”. Please refer to the modified details in the revised version.

Point 16: In section 3.5, the specific drug tested by the author is not specified. Additionally, the author repeats information that has already been provided in section 3.1.

Response 16: Thank you for your suggestion. D1 and fluopyram mentioned in Section 3.5 are the specific drugs tested. In addition, repeated information was deleted. Please refer to the modified details in the revised version.

Point 17: The author's description of the docking of fluopyram with SDH and D1 may confuse the reader, as it is incorrect or misleading.

Response 17: Thank you for your suggestion. The molecular docking part has been modified. Please refer to the modification details of the 2.3.5 part in the revised version.

Point 18: The author should clearly indicate the solution used to treat the mycelium and provide an explanation for using two separate frequencies to measure the optical densities. 

Response 18: Thank you for your suggestion. The solution used to treat the mycelium in section 3.8  is sterile water. The reason for choosing 600 and 532 nm to measure the optical densities is that malondialdehyde can react with thiobarbituric acid (TBA) to form reddish-brown trimethyldione under acidic and high temperature conditions. It has the maximum light absorption at 532 nm and the minimum light absorption at 600 nm. Please refer to the modified details in the revised version.

Point 19: The author did not explain how they calculated the EC50 values from the optical densities. 

Response 19: Thank you for your suggestion. In section 3.2, the calculation method of EC50 value is supplemented, specifically “According to the above method, the inhibition rates of 100, 50, 25, 12.5, 6.25 μg/mL were tested. The inhibition rate was converted into probability value (y), and the concentration of the drug solution was converted into logarithmic value (x). The data were processed by Excel software to obtain the virulence regression equation (y = ax+b) and correlation coefficient (R), so as to calculate the half maximal effective concentration (EC50) of the excellent compound against the pathogen.” Please refer to the modified details in the revised version.

Point 20: The author did not mention the specific mechanistic studies they conducted.

Response 20: Thank you for your suggestion. The specific mechanisms mentioned in this paper are : scanning electron microscope and microscope imaging test, SDH enzyme activity inhibition test, molecular docking, malondialdehyde content determination. Please refer to the modified details in the revised version.

Point 21: The conclusion section provides a brief summary of the results rather than offering an overall inference or conclusion drawn from the study.

Response 21: Thank you for your suggestion. In the conclusion section, the experimental results are briefly summarized. Please refer to the modified details in the revised version.

Comments on the Quality of English Language

Point 1: There are multiple sentences that are redundant and challenging to comprehend due to poor writing, such as in line numbers 32, 42-43, 56-57, 103, 122-123, 128-129, 144-147, 237-238, 289-291, 319-321.

Response 1: Thank you for your suggestion. Professionals have been invited to modify the entire article. Please refer to the modified details in the revised version.

Point 2: The author's choice of words appears superficial, non-scientific, and inappropriate in the given context. Several words such as "better inhibition," "novel mechanism" (without detailed explanation), "hot structure," "good biological activity," "better than," "better inhibition," "general antifungal activity," "certain antifungal activity," "normal mycelium," "somewhat inhibited," "precipitation dried by filtration," "react overnight," "general procedure," "until complete," "96 well-plate method," "bacterial culture" (not bacterial solution), "three replicate (instead triplicate)," "on SEM stage" (instead of SEM stub), "sprayed with gold" (instead of sputter coated with gold), "on SEM" (instead of under SEM), "growth rate method," "dried with filter paper" (instead of wiped with filter paper), "perforated with punch" (instead of punched holes), "cultured" (instead of incubated), "fully ground," "cell membrane damage," "poor solubility problem," "against kiwifruit," "folded and curled" should be reconsidered

Response 2: Thank you for your suggestion. The superficial, non-scientific and inappropriate words in the text were modified. Please refer to the modified details in the revised version.

Point 3: The author's description of the docking of fluopyram with SDH and D1 may confuse the reader, as it is incorrect and misleading.

Response 3: Thank you for your suggestion. The molecular docking part has been modified. Please refer to the modified details in the revised version.

Point 4: Random capital letters have been utilized in the text, for example, in line 29.

Response 4: Thank you for your suggestion. The random capital letters used in the text have been modified. Please refer to the modified details in the revised version.

Round 2

Reviewer 3 Report

Thank you for addressing the suggested scientific concerns. Unfortunately, none of the scientific concerns have been resolved. Please refer to the following comments regarding your responses to my major concerns:

 1.       The introduction section should emphasize the scientific correlation between SDH, cell damage, and synthesized myricetin derivatives, with appropriate references.

 2.       Instead of focusing on the results obtained, the procedure to synthesize compounds D1 to D19 is mentioned in the results and discussion section.

 3.       Line 143 indicates that the scanning electron microscopy study did not fulfill the scientific concern regarding the mechanism of action at the molecular level.

 4.       Physicochemical properties data for both the control and test compounds, as well as high-resolution SEM images, are missing.

 5.       I remain unconvinced by the authors' argument. Conducting experiments solely on picked fruits rather than the entire plants in the soil qualifies as in vitro experiments.

 6.       The authors have not thoroughly investigated molecular interactions to establish the relationship between myricetin derivatives and SDH, suggesting a bias in their observations.

 7.       The authors failed to provide a reference for the MDA test hypothesis or establish a scientific connection between membrane damage and SDH activity.

Thanks for showing your sincere efforts and working with experts in the English language to evaluate the written language of the manuscript. However, some concerns are not yet addressed like non-scientific terms like Hot Structure instead of popular compound/structure (Hot is representing temperature.), good bactericidal and antifungal activities instead of signification activities.